# How to Assess Pulmonary Circulation and Right Heart Chambers in Systemic Sclerosis Patients?

**DOI:** 10.3390/diagnostics15081029

**Published:** 2025-04-17

**Authors:** Michele Correale, Ester Maria Lucia Bevere, Lucia Tricarico, Deborah Villani, Mattia Granato, Erminia Guerriero, Raffaele Capasso, Luciano Rossi, Cinzia Rotondo, Francesco Paolo Cantatore, Addolorata Corrado, Massimo Iacoviello, Natale Daniele Brunetti

**Affiliations:** 1Cardiothoracic Department, Ospedali Riuniti University Hospital, 71100 Foggia, Italy; 2Department of Medical and Surgical Sciences, University of Foggia, 71122 Foggia, Italy; estermarialuciabevere@gmail.com (E.M.L.B.); lucia.tricarico.lt@gmail.com (L.T.); villani.deborah89@gmail.com (D.V.); mattia.granato@unifg.it (M.G.); erminia.guerriero.eg@gmail.com (E.G.); raffaelecapasso@outlook.com (R.C.); luciano.rossi17@gmail.com (L.R.); massimo.iacoviello@gmail.com (M.I.); natale.brunetti@unifg.it (N.D.B.); 3Rheumatology Clinic, Department of Medical and Surgical Sciences, University of Foggia, 71122 Foggia, Italy; cinzia.rotondo@gmail.com (C.R.); francescopaolo.cantatore@unifg.it (F.P.C.); addolorata.corrado@unifg.it (A.C.)

**Keywords:** systemic sclerosis, heart failure, PAH, pulmonary arterial hypertension, pulmonary circulation, pulmonary hypertension

## Abstract

Systemic sclerosis (SSc) is a rare autoimmune connective tissue disease characterized by a widespread accumulation of extracellular matrix components leading to fibrosis of the skin and internal organs. Vascular changes occur in all involved tissues and are responsible for several distinctive clinical manifestations of the disease. This review focuses on the usefulness of various diagnostic tools in clinical practice for the early identification of clinical, functional, and/or structural RV impairment in SSc patients at risk of PH. It aims to identify specific causes of RV dysfunction, describe potential differences in outcome measures, and, ultimately, determine different cut-off values compared to subjects with PH not related to SSc.

## 1. Introduction

Systemic sclerosis (SSc) is a rare autoimmune connective tissue disease characterized by a widespread accumulation of extracellular matrix components leading to fibrosis of the skin and internal organs, such as lung, gastrointestinal tract, heart and kidney, and *vasculopathic* changes that affect both micro- and macrovasculature [1]. Vascular involvement represents a central aspect of the pathogenesis of the disease and consists of the loss of small vessels associated with obliterative vasculopathy affecting small arteries, arterioles, and capillaries. Vascular changes occur in all involved tissues [2] and are responsible for several distinctive clinical manifestations of the disease, particularly Raynaud phenomenon, digital ulcers, telangiectasias scleroderma renal crisis, gastric antral vascular ectasia, and pulmonary arterial hypertension (PAH) [3]. Structural changes in the vascular bed led to increased pressures in the pulmonary circulation, severely impacting the right heart and significantly affecting mortality. Furthermore, differences in the innate response of the right ventricle to increased pulmonary vascular resistance have been reported in SSc patients. PAH is a severe complication of SSc, determining a significant impact on morbidity and quality of life, and represents a leading cause of mortality in patients affected by this disease [4]. PAH is a clinical entity characterized hemodynamically by the presence of pre-capillary pulmonary hypertension (PH), occurring without other known causes of pre-capillary PH, such as lung fibrosis and pulmonary veno-occlusive disease [5]. Due to the complex pathogenesis of the disease and the widespread fibrotic and vascular abnormalities affecting multiple organs and tissues, SSc differs from idiopathic PAH (IPAH) in that it can present with various forms of pre-capillary PH as well as post-capillary causes of PH. In SSc, PH can be associated with pulmonary arterial vasculopathy (SSc-PAH) or pulmonary veno-occlusive disease (group 1 PH), left heart involvement (group 2), pulmonary fibrosis (group 3), or chronic thromboembolic disease (group 4) [6]. These distinct phenotypes frequently exhibit a significant pathophysiological overlap, making the management of these patients particularly challenging. PH represents a main cause of critical impairment of right ventricle (RV) function in SSc patients; nevertheless, RV impairment can be observed regardless of PH due to direct damage induced by the disease itself, which can involve the myocardium, the conduction system, the valves, the pericardium and the conduction system [7].

Managing pulmonary vascular complications in SSc requires a multidisciplinary team (MDT) approach [8]. Collaborative decision-making between rheumatologists, pulmonologists, cardiologists, and radiologists ensures comprehensive assessment and optimized treatment strategies. Regular MDT meetings allow for case discussions, interpretation of complex imaging findings, and coordination of therapeutic interventions. This approach improves patient outcomes by fostering early diagnosis and timely intervention.

We propose a stepwise diagnostic algorithm for SSc patients at risk of PH, which arises from the MDT approach **(**Figure 1). We report possible deviation for SSc from the usual PAH assessment in Table 1.

This review focuses on the usefulness of various diagnostic tools in clinical practice for the early identification of clinical, functional, and/or structural RV impairment in SSc patients at risk of PH (Figure 2). It aims to identify specific causes of RV dysfunction, describe potential differences in outcome measures, and, ultimately, determine different cut-off values compared to subjects with PH not related to SSc.

## 2. Imaging Methods on the Right Heart and Pulmonary Circulation

Echocardiography (Figure 3).

### 2.1. Two-Dimensional Echocardiographic and Doppler Measurements

Transthoracic echocardiography (TTE) is a non-invasive tool for pulmonary hypertension (PH) screening and follow-up. The ESC and ERS recommend PH screening in high-risk patients, such as those with systemic sclerosis (SSc), where PH prevalence is 5–19% [9] and a leading cause of death. In suspected PH cases based on symptoms and clinical findings, early detection enables timely treatment and better outcomes.

An estimation of pulmonary artery systolic pressure (PAsP) can be provided, rather than an exact measured value [10]. PAsP is based on systolic peak tricuspid regurgitation velocity TRV and the estimated right atrial pressure (RAP). RAP is typically estimated by assessing the diameter of the inferior vena cava (IVC) and its percentage of collapse during inspiration [11].

RAP estimation has limited accuracy, so ESC guidelines recommend using systolic peak TRV as an indirect indicator for assessing the probability of PH. A peak VRT > 3.4 m/s identifies a high probability of PH.

A peak TRV > 2.8 m/s may suggest it [5]. In this case, additional two-dimensional echocardiographic and Doppler measurements are used to define the echocardiographic probability of PH, which may then be determined as low, intermediate, or high [5]. These parameters are right ventricle (RV)/left ventricle (LV) basal diameter ratio > 1.0; flattened interventricular septum leading to “D-shape” LV (LV eccentricity index > 1.1); RV outflow tract Doppler acceleration time (RVOT AT) < 105 ms and/or mid-systolic notching; early diastolic pulmonary regurgitation velocity > 2.2 m/s; inferior vena cava diameter > 21 mm with decreased inspiratory collapse; a right atrial (RA) area at end-systole >18 cm^2^ [5].

Other indirect echocardiographic signs of PH are a pulmonary artery to aortic root ratio > 1 and/or PA diameter > 25 mm; decreased Tricuspid Annular Plane Systolic Excursion (TAPSE) < 18 mm; decreased peak systolic velocity of tricuspid (S’ wave) < 9.5 cm/s; reduced right ventricular fractional area change (RV-FAC) < 35%; TAPSE/PAsP ratio < 0.55 mm/mmHg [11]. Signs from at least two categories (pulmonary artery, RV, right atrium/inferior vena cava) increase PH’s echocardiographic probability level.

A cardiological work-up, including TTE, must be performed periodically in SSc patients. Indeed, RV echocardiographic parameters are not infrequently abnormal in these patients, with or without PH, compared to the healthy population [12].

Right ventricular dysfunction was traditionally believed to be a consequence of increased pulmonary arterial pressure. Accumulating evidence highlights a direct involvement of systemic sclerosis (SSc) in the remodeling and dysfunction of the right ventricle (RV), with right ventricular impairment being significantly more marked in SSc-associated pulmonary arterial hypertension (PAH) compared to non-SSc PAH [13].

Durmus et al. observed a significant reduction in TAPSE and significantly higher PAsP in SSc patients [12,14]. In patients with SSc-PAH, alterations in eccentricity index, TAPSE, RVOT ACT, and TRV are more frequent compared to those with SSc without PAH [12]. Further studies are needed to identify new echocardiographic characteristics in PH-SSc patients (Table 2).

### 2.2. Three-Dimensional Echocardiography

Advances in three-dimensional echocardiography (3DE) allow for better detection of RV abnormalities and early signs of PH, offering more accurate volume measurement and RVEF estimation than two-dimensional imaging [13,15,16]. Few studies have assessed 3DE in SSc patients. Pigatto et al. found that SSc patients, even without clinical heart disease, had larger RV volumes, reduced 3DRVEF, and higher TRV peaks compared to healthy controls [13].

Tadic et al. showed similar results demonstrating decreased 3DE-RVEF (50% ± 6 versus control 59% ± 5) and higher RV end-systolic volume (48 mL ± 13 versus control 35 mL ± 9). However, RV end-diastolic volume and stroke volume did not significantly differ between the SSc group and healthy controls [17].

Studies indicate mild RV abnormalities, suggesting early impairment but not confirming PH. Additional parameters like RA area and TRV peak (DETECT algorithm) may enhance evaluation [18]. Anyway, 3DE limitations consist of load state dependency and complexity in acquiring suitable images of the RV chamber [19]. Among the parameters of interest, Ventricular–Arterial Coupling (VAC) captures the dynamic interplay between cardiac performance and arterial properties and is recognized as a major determinant of global cardiovascular efficiency. VAC is defined as the ratio of arterial elastance (Ea) to left ventricular (LV) end-systolic elastance (Ees). Tona et al. pioneered the investigation of the relationship between LV pressure–volume characteristics and VAC using three-dimensional echocardiography (3DE) in systemic sclerosis (SSc) patients [20] (Table 1).

### 2.3. Tissue Doppler Imaging (TDI)

Pulsed tissue Doppler echocardiography (TDI) is a key tool for assessing ventricular contractility, with a tricuspid annular systolic velocity (S’ wave) <9.5 cm/s indicating RV impairment [21]. SSc patients often develop myocardial fibrosis, leading to diastolic dysfunction. Meune et al. found that 15% of SSc patients had reduced RV contractility, 14% had reduced LV contractility, and LV diastolic dysfunction was present in 17–30% compared to healthy controls [22].

Another indicator of diastolic and systolic functions is the RV myocardial performance index (RVMPI) or RV Tei index [23]. By using PW-TDI measurements from the lateral tricuspid annulus, it can be calculated as the ratio of isovolumic contraction (ICT) and relaxation time (IRT) intervals to ventricular ejection time (ET) [24]. Vonk et al. observed a strong correlation between RVMPI and mPAP in SSc patients. In their study, 19 of 20 patients with RVMPI > 0.36 had PAH confirmed by RHC, though 9 of 15 with RVMPI ≤ 0.36 also had PAH [25] (Table 1).

### 2.4. Right Atrial and Ventricular Strain

Standard echocardiography detects RV dysfunction and PH only in symptomatic patients, with limited ability to identify early abnormalities. Speckle-tracking echocardiography assesses myocardial deformation during contraction and relaxation. Free wall RV longitudinal strain (RVLS) is preferred over global strain, as it excludes the interventricular septum, which is more influenced by left ventricular function.

In a case–control, single-center study, left and right ventricular GLS were significantly impaired in patients with SSc when compared to controls, and among SSc patients, it did not differ according to ANA positivity [26].

Mukherjee et al. studied the RV strain of SSc patients compared with a control group. There were no significant differences in 2D Echocardiographic parameters of RV chamber size or function but patients with SSc had diminished RV longitudinal strain (RVLSS) compared with control consistent with RV dysfunction. Comparing the regional abnormalities, they found a specific pattern. RVLS decreased in the apex and mid-segment while it increased in the base. Basal hyperkinesis combined with mid and apical hypokinesis was consistently detected in systemic sclerosis (SSc) patients, independent of right ventricular systolic pressure (RVSP), even among those with RVSP < 35 mmHg. These findings suggest underlying myocardial disease that may manifest prior to the clinical development of pulmonary arterial hypertension (PAH) [27].

Another investigation aimed to elucidate the relationship between pulmonary fibrosis, elevated pulmonary pressures (PHT), and right ventricular (RV) function in systemic sclerosis (SSc) patients, utilizing thoracic computed tomography (CT) and speckle tracking-derived assessment of RV-free wall strain. The study demonstrated that individuals with both pulmonary fibrosis and PHT exhibited the greatest impairment in RV-free wall strain compared to those with either isolated pulmonary fibrosis or no evidence of fibrosis or PHT. Furthermore, multivariate regression analysis revealed pulmonary fibrosis and left ventricular ejection fraction as independent correlates of compromised RV function [28] (Table 1).

### 2.5. RV-PA Coupling

RV-PA coupling reflects the balance between RV contractility and afterload, which is crucial for maintaining cardiac function. When afterload increases, preserved coupling relies on RV hypertrophy to sustain cardiac index (CI), TAPSE, and RVEF. However, RV-PA decoupling occurs when contractility fails to compensate, leading to RV dysfunction and failure. Therefore, early evaluation is essential for risk stratification and prognosis. The assessment of RV-PA coupling utilizes both invasive techniques, such as RHC, and non-invasive techniques, such as CMR. RHC is considered the gold standard for estimating RV-PA coupling, as it allows the evaluation of the relationship between multi-beat end-systolic elastance (Ees) and effective arterial elastance (Ea) through the analysis of a pressure-volume curve [29].

The TAPSE/PASP ratio is the most accurate non-invasive method for estimating RV-PA coupling and serves as a key prognostic marker [30]. TAPSE assesses RV contractility, while PASP estimates RV afterload, making their ratio a viable alternative to the Ees/Ea ratio from RHC. A TAPSE/PASP ratio below 0.36 mm/mmHg indicates significant RV-PA impairment. However, its accuracy is limited in cases of severe TR or regional RV kinetic abnormalities [29]. Despite the undefined exact correlation with Ees/Ea, TAPSE/PASP reliably stages disease severity and predicts prognosis, highlighting RV contractility as the primary determinant of PH outcomes. Its inverse relationship with the NYHA class enables early symptom detection, even in mild cases, reinforcing its role as a reliable surrogate for Ees/Ea [31] (Table 1).

### 2.6. Biomarkers and Imaging for Risk Stratification

NT-proBNP is a key biomarker used in assessing right heart strain and potential PH in systemic SSc patients [32]. Elevated levels correlate with right ventricular dysfunction and higher pulmonary pressures. Integrating NT-proBNP measurements with imaging modalities such as echocardiography and CMR can improve risk stratification. Studies suggest that combining NT-proBNP with echocardiographic parameters, such as TRV and right atrial area, enhances diagnostic accuracy [33]. This integration allows for early identification of patients who may benefit from invasive confirmation via RHC.

The DETECT algorithm further refines risk stratification in SSc patients at risk for PAH [18]. This evidence-based tool incorporates clinical, serological, and echocardiographic variables—including NT-proBNP, forced vital capacity (FVC)/diffusing capacity for carbon monoxide (DLCO) ratio, and right atrial area—to identify patients requiring RHC. The DETECT algorithm has demonstrated superior sensitivity in detecting PAH compared to traditional echocardiographic thresholds alone, reducing missed diagnoses in early-stage disease.

The authors concluded that NT-proBNP is a useful tool for early screening and diagnosis but cannot be used alone in these SSc-PAH detection strategies; it should be integrated into a more comprehensive evaluation. Additionally, as a future direction, the authors suggest evaluating new serum biomarkers to be used alongside NT-proBNP.

## 3. Cardiac Magnetic Resonance (CMR) Imaging

Cardiac MR (CMR) is a valuable, non-invasive diagnostic tool that provides high-resolution 3D images of the myocardium, enabling comprehensive structural, volumetric, and functional evaluation of the right heart and its changes over time. This capability is difficult to achieve with other techniques, such as echocardiography and RHC. CMR is particularly effective in evaluating PAH by assessing the right ventricle and its relationship with the left ventricle and identifying any damage to the right ventricle. CMR provides detailed information about anatomy, perfusion, function, vitality, and hemodynamic and metabolic variations of the heart–lung system. By encoding the CMR signal, it is possible to calculate phase volumetric flow, which is suitable for determining various parameters such as end-diastolic volume, end-systolic volume, ejection fraction (EF), ejection volume, and cardiac output (CO). However, CMR is less effective in estimating pulmonary artery pressure (PAP) and pulmonary vascular resistance (PVR) compared to RHC [34]. Key prognostic markers identified through CMR include right ventricular end-diastolic volume (RVEDV), right ventricular stroke volume (RVSV), and the curvature of the interventricular septum. Phase contrast MR (PC-MR) is used to assess various parameters of the pulmonary artery (PA), including flow variations, average flow velocity, relative area change (RAC), wedge pressure velocity, and elasticity of the pa. The first three parameters are particularly useful for evaluating a patient’s compliance with pharmacological treatments [35]. In SSc, myocardial perfusion deficit can be caused by CAD or non-atherosclerotic coronary insult. Nowadays, SPECT/CT with Tc-99m-Sestamibi allows the diagnosis of myocardial perfusion deficit in 88% of symptomatic patients, as well as SPECT/CT with Ta-201-iobenguane has been shown to be equally effective. However, these techniques have long execution times and a high exposure to ionizing radiation, while CMR is free from limitations. Kobayashi et al. demonstrated that steady-state free flow (SSFP) CMR was equally reliable as LGE, which is commonly used in the evaluation of the degree of myocardial replacement fibrosis. Furthermore, it was seen that the presence of Raynaud’s phenomenon in the previous 15 years was associated with areas of LGE of greater diameter. Since SSFP does not require contrast agents, it may offer a safer and more accessible alternative for patients with contraindications to gadolinium, such as those with impaired renal function. Additionally, the observed association between Raynaud’s phenomenon and larger areas of LGE suggests that a history of prolonged Raynaud’s could serve as an early clinical marker for myocardial involvement in SSc. Nicholson et al. finally demonstrated a direct proportionality between LGE and the extent of pulmonary fibrosis and those with an impairment of RVEF [35].

Recent PAH research focuses on improving non-invasive diagnostics, with CMR emerging as a promising tool for detecting pulmonary circulation changes before RV involvement. PAH patients show reduced PA peak velocities, correlating with higher PASP and PVR. Studies reveal significantly lower PA elasticity in PAH patients versus healthy individuals, indicating impaired vascular response to stress [36,37]. Evaluating cardiac output(CO) and RVSV is crucial, as Oberbeek et al. found PAH patients with SSc have lower RVSV than those with idiopathic PAH, linking reduced contractility to poorer exercise tolerance, higher morbidity, and increased mortality [22,38].

Research indicates that increased right atrial volume, assessed by CMR, is a poor prognostic factor in PAH. While CMR and echocardiography are equally effective for PAH diagnosis and prognosis, CT is less reliable [37,39]. Echocardiographic assessment of the RV in PH patients is challenging due to myocardial remodeling and septal motion abnormalities [35,37]. In contrast, CMR provides highly accurate measurements of ventricular volumes, mass, and kinetic disorders using balanced SSFP sequences with ECG gating. CMR outperforms echocardiography, which underestimates RV volumes, and CT, which overestimates them. Notably, indexed RVEDV is the strongest predictor of mortality in PH [35,40].

A study by Knight et al. identified five CMR-based cardiac phenotypes linked to varying mortality rates. In SSc-PAH, 45% of patients fell into ‘RV failure’ or ‘normal function, large cavity’ groups, both associated with elevated ECV and worse prognosis [41]. Knight et al.’s study has important clinical implications for the management of SSc-PAH. By identifying cardiac phenotypes linked to different mortality rates, their findings can aid in risk stratification, allowing clinicians to better predict which patients are at higher risk of poor outcomes. Patients classified into the ‘RV failure’ or ‘normal function, large cavity’ groups, both of which exhibited elevated extracellular volume in T1 mapping, may require closer monitoring and more aggressive treatment strategies, such as early initiation of pulmonary vasodilators or cardioprotective therapies. Additionally, incorporating CMR-based phenotyping into clinical practice could refine existing treatment guidelines and improve patient counseling by providing a clearer understanding of disease progression and prognosis. CMR has also refined ventricular mass index (VMI) estimation, with a VMI < 0.7 linked to a 91% two-year survival, while >0.7 indicates poorer outcomes. Native T1 is the strongest predictor of cardiovascular mortality in SSc, independent of skin involvement [42]. While RV volumes are prognostic in PH, native T1 provides additional independent prognostic value in SSc-PH.

Some studies have also demonstrated that native myocardial T1 remains a prognostic marker in SSc, even in the absence of PH diagnosis, suggesting that the increased native T1 in SSc-PH is more likely related to myocardial involvement caused by SSc rather than to the effects of PH. This distinction may influence treatment approaches [43]. Advances in cardiac imaging are promoting the increasing use of CMR for diagnosing and dynamically evaluating PAH [37] (Table 1).

## 4. Exercise Testing

Exercise tests are important tools in the management of patients affected by SSc. They are used for screening and early detection of PAH, stratifying risk, monitoring disease progression, and assessing treatment efficacy. These instruments are capable of detecting early forms of PH that are evident only during exercise [44]. The pulmonary circulation has meaningful microcirculatory reserves at rest. These reserves are mobilized during exercise, leading to an expansion of the capillary surface area for efficient gas exchange. PAP remains low despite this increased blood flow. Any decline in these vascular reserves may become evident during physical activity [45]. While RHC remains the definitive diagnostic method, stress tests can help identify patients who really need this invasive procedure, minimizing risks and costs [44].

### 4.1. Incremental Shuttle Walking Test vs. Six Minutes Walking Test

The Incremental Shuttle Walking Test (ISWT) is a standardized, externally paced field walking test designed to assess functional capacity in patients affected by chronic heart failure, PH, and other chronic lung diseases. It provides valuable information about functional capacity and correlates strongly with VO2 max during cardio-pulmonary exercise test (CPET) [46]. It reflects disease severity in patients affected by pulmonary hypertension. In a study conducted by Billing et al., the ISWT was correlated with WHO functional class stronger than the Diffusing Capacity of the Lungs for Carbon Monoxide (DLCO). A relevant correlation has also been demonstrated with some hemodynamic parameters such as mean PAP, systolic PAP, PVR, mRAP, CI, and mixed venous oxygen saturation (SmVO2). Patients with a new diagnosis of PH with no or mild symptoms (WHO I), normal diffusion of carbon monoxide, and modest PAP show a significant reduction of exercise capacity during this test. Compared to patients in WHO FC I, patients in WHO FC II had a lower exercise capacity. Therefore, patients without symptoms may experience a reduction in exercise capacity and an elevation of PAP during RHC. This suggests ISWT could serve as an early screening tool for PH, particularly in SSc patients, aiding in the selection of RHC and improving disease management [47].

The Six-Minute Walking Test (6MWT) is a standardized, reliable measure of sub-maximal exercise capacity, assessing the distance a patient can walk in six minutes. Originally developed for cardiopulmonary conditions like PH and heart failure [48], it is also used in SSc, though not fully validated. In SSc patients, the 6MWT is valuable for early diagnosis, risk stratification, therapy selection, and treatment monitoring [49,50]. Studies have shown that SSc patients with PAH walk less during the 6MWT than those without PAH. Similarly, SSc patients with interstitial lung disease (ILD) and PH exhibit reduced walking distances compared to SSc patients without ILD or PH [51,52]. SSc patients can develop conditions of group 1 PH (precapillary PH, PAH, and veno-occlusive disease), group 2 PH (post-capillary PH due to left ventricular diastolic dysfunction), group 3 PH (pre-capillary PH due to ILD), and group 4 (chronic thromboembolic disease). ILD is a very common clinical manifestation in SSc; PH in these patients can be an isolated manifestation of the disease or may be associated with ILD. Thus, all SSc patients should undergo a comprehensive clinical assessment, including lung function tests and whole-body plethysmography, along with lung imaging by High-Resolution Computed Tomography (HRTC) [53].

Lung function tests in SSc patients typically show a reduced forced vital capacity (FVC) and a reduction of the diffusion capacity of the lung for carbon monoxide (DLco); nevertheless, a reduction of DLco is observed also in PH, so in these patients, it is mandatory to interpret the DLco within the whole clinical contest, considering, in particular, the alterations of FVC compared to DLco variations (FVC/DLCO ratio) and the extent of interstitial lung changes [54].

Some studies have demonstrated alterations in right ventricular contractility in patients with SSc. Right heart alterations may be in SSc patients and PAH and those that occur without PAH, and that can be attributed to other causes such as microvascular damage or myocardial fibrosis [55].

Decreased exercise capacity significantly correlates with echocardiographic parameters of RV dysfunction, NT-proBNP, and endothelin-1 levels in SSc patients [56]. Some parameters indicative of left ventricular diastolic function and right ventricular systolic function, such as left atrial volume index and right FAC, can predict the worsening of 6MWT distance in patients affected by SSc [57,58,59]. There are also correlations between hemodynamic parameters and the distance covered [57]. In a study by Kovacs et al., mean pulmonary artery pressure (mPAP) and pulmonary vascular resistance (PVR) at the higher end of the normal range at rest, along with moderate increases during exercise, were associated with impaired exercise tolerance and may represent early manifestations of pulmonary vasculopathy in systemic sclerosis (SSc) patients [60].

The 6MWT reflects daily physical activity levels, which are reduced in PAH patients and correlate with walking distance. However, it cannot be used alone for PH screening due to factors like osteo-articular involvement. In pulmonary involvement, it should be combined with the Scleroderma Health Assessment Questionnaire Disability Index, DLCO, and its components to predict PH onset [61,62,63,64]. The 6MWT is useful for monitoring disease progression [65] and treatment response, with studies showing improved walking distances in patients treated with Bosentan or Treprostinil [66,67,68].

Saturation evaluation is crucial for patient monitoring, as SaO2 remains more stable in SSc patients without PAH compared to those with SSc-PAH. A reduction in SaO2 during exercise enhances the predictive value of the 6MWT [69] (Table 1). The ISWT may be preferable for assessing symptom-limited exercise capacity in SSc patients due to its standardization, reproducibility, and stronger correlation with VO2 peak and VO2 at AT, eliciting higher cardiopulmonary responses [70]. Nocturnal pulse oximetry is not a diagnostic tool used in current clinical practice for the assessment of cardiopulmonary disease in SSc patients unless risk factors or Obstructive Sleep Apnea Syndrome (OSAS) are present, as in the general population. A recently published pilot study evaluating the full night polysomnography parameters, performed on a very limited cohort of SSc patients, showed that the apnea–hypopnea score and duration of SpO2 were higher in subjects with PH compared to subjects without PH, but the difference was not significant [54].

### 4.2. Cardiopulmonary Exercise Test (CPET)

The cardiopulmonary exercise test (CPET) is a useful method in the comprehensive assessment of pulmonary vascular function in SSc patients. CPET provides an integrative and non-invasive approach to assess the functional capacity of the cardiopulmonary system, showing the hemodynamic and gas exchange abnormalities that characterize pulmonary vascular disease in SSc. CPET involves the measurement of various physiological parameters. These variables include oxygen uptake (VO2), carbon dioxide production (VCO2), minute ventilation (VE), and heart rate (HR). These parameters are used to derive key indices such as the ventilatory equivalent for carbon dioxide (VE/VCO2 slope) and the oxygen uptake efficiency slope (OUES), which reflect the efficiency of gas exchange and ventilatory control [71].

CPET is important for both diagnosis and prognosis [8,72]. It allows us to understand the cause of dyspnea and reduced exercise tolerance as well as to exclude the presence of PH [73,74,75]. Studies have demonstrated that SSc patients with pulmonary vascular disease exhibit distinct CPET profiles characterized by reduced peak VO2, elevated VE/VCO2 slope, and impaired OUES. VEVCO2 is a hallmark characteristic of pulmonary vascular diseases [76]. Combining increased VE/VCO2 slope with reduced PETCO2 aids in the diagnosis of pulmonary vasculopathy. VO2max, VO2AT, and O2 pulse were significantly correlated to baseline PAsP [77]. PAH patients exhibit lower OUES plateau and slope, indicating ventilatory inefficiency. A reduction of work rate (WR), VO2/WR relationship, and VD (dead volume)/VT (tidal volume) ratio are also common in PAH patients. Blood gas analysis during exercise is crucial for accurate VD/VT assessment. VD/VT increased to > 30% indicates pulmonary vascular limitation [69].

CPET could be used for PAH screening in SSc patients without pulmonary or cardiac disease because it can help identify early signs of pulmonary vascular dysfunction [44,78,79], even in the absence of resting hemodynamic abnormalities and normal pulmonary function tests [80]. In SSc patients with normal PAP at rest, a decreased oxygen uptake might be caused by sudden PAP increases during exercise. These patients should be considered at risk for developing pulmonary hypertension [81]. Kanazawa et al. have shown that exercise PH may represent an intermediate condition between not having PH and overt PH. In fact, these patients present intermediate values of peak VO2 and VEVCO2 slope [82]. SSc patients without pulmonary involvement present reduced VO2 peak, reduced Metabolic Equivalent of Task (METS) at peak exercise, and a shorter time interval between anaerobic threshold (AT) and respiratory compensation point (RCP), probably related to an abnormal vascular response to exercise [83]. A low pulse O2 and low VO2 at the anaerobic threshold are compatible with pulmonary vasculopathy [84].

The abnormalities detected at CPET correlate with the severity of PH and RV dysfunction. Several exercise variables (peak VO2, VE/VCO2 slope, oxygen saturation—SaO2) obtained at the beginning of follow-up are predictive of adverse events. Reduction in VO2 peak and increase in VEVCO2 are prognostic markers in patients affected by SSc [85,86]. An increase in VEVCO2 appears to correlate with an increase in PAP [87], with the activity index and the severity of the pathology [78]. VEVCO2 may improve with interventions [76]. In a study conducted by Cuomo et al., VO2 peak was found to be associated with the extent of pulmonary and cardiac involvement, with the left ventricular diastolic dysfunction and the Health Assessment Questionnaire-Disability Index (HAQ-DI) score [88]. Another prognostic indicator is the O2 pulse [89]. Badagliacca et al. found that oxygen pulse and RV systolic function enhance risk prediction models. Low RV FAC and oxygen pulse combination may identify high-risk patients [90]. CPET parameters may predict death and pulmonary complications [91]. Ventilatory efficiency assessment identifies patients at increased risk of death [69]. According to the 2022 ESC/ERS guidelines on PH, in symptomatic patients affected by SSc, CPET may be used to decide whether or not to perform RHC as an alternative to stress echocardiography and CMR (class IIb) [5]. CPET can aid in differentiating between pulmonary vascular disease and other causes of exercise limitation. Impaired exercise capacity can also be associated with diffuse myocardial fibroinflammatory disease identified at CMR and skeletal muscle edema [92]. The combination of CPET values with blood serology, lung function test, thorax high-resolution computer tomography, and Doppler echocardiogram could help to discriminate the causes of exercise limitation. The values of ΔPETC02, peak VO2 (% predicted), peak heart rate (HR% predicted), peak WR (% predicted), and AT (% predicted) could distinguish pulmonary vasculopathy from other causes of exercise limitation [93]. According to Demitrescu et al., both patients with left ventricular dysfunction and pulmonary vasculopathy had low VO2 at peak and AT, but a higher VE/VCO2 at AT and decreasing PETCO2 during early exercise distinguished pulmonary vasculopathy from the other form [94,95]. Martis et al. affirm that VEVCO2 values alone are insufficient for determining the etiology of exercise limitation. According to these authors, increased dead space suggested pulmonary vascular disease in the absence of a high respiratory rate. A low oxygen pulse value indicated left ventricular dysfunction [96]. An important role could be played by CPET with RHC (CPET/RHC). The exercise alveolar–arterial oxygen gradient [P(Ai-a)O2] was elevated in SSc patients with pulmonary vasculopathy compared to those with exercise left ventricular diastolic dysfunction and deconditioning [97]. Only the [P(Ai-a)O2] and DLCO helped distinguish patients without ILD compared with those affected by ILD [96]. Patients with ILD exhibit a reduction in exercise capacity compared to those without ILD [98]. In these patients, VE/VCO2 slope is a prognostic marker of five-year mortality for ILD and major vascular complications [99]. CPET parameters have a significant correlation with pulmonary hemodynamics. Peak VO2 and VE/VCO2 correlated with PAP, Transpulmonary Pressure Gradient (TPG), and PVR [100]. VE/VCO2 slope is the best parameter able to predict PAH at RHC. It was directly related to the Diastolic Pressure Gradient (DPG), TPG, mean PAP, and PVR. An increase in ventilatory equivalents of CO2 and O2 and a reduction in PETCO2 seem to be associated with hemodynamic impairment. The combination of peak PETCO2 and VE/VCO2 slope helps to stratify patients according to mPAP, PVR, CI, and RA pressure (RAP) values. Patients with VE/VCO2 slope above the median value and PETCO2 under the median value more frequently show lower CI and higher afterload and filling pressure [89]. Another parameter is ventilatory power (VP). It presents significant correlations with DPG, TPG, mPAP, and PVR. Correlations with DPG and PVR might help in discerning isolated post-capillary PH from combined post-capillary and pre-capillary PH [101]. Also, the intercept of ventilation (VEint) during exercise might predict the results of RHC. VEint and VE/VCO2 slope are inversely correlated to DPG, TPG, mPAP, and PVR. In addition, VEint and VE/VCO2 slope could be helpful in differentiating isolated postcapillary PH from combined PH [100].

So, these CPET parameters could be used to choose patients to undergo invasive procedures, minimizing risks and costs [102], in addition to other techniques [103].

In patients with SSc-PAH, NT-proBNP levels showed no significant correlations with CPET parameters compared to idiopathic PAH (IPAH) and chronic thromboembolic pulmonary hypertension (CTEPH) [104]. Peak VO2 reflects functional impairment and correlates with NYHA class [65].

In a study conducted by Brown et al., cardiovascular magnetic resonance—augmented cardiopulmonary exercise testing (CMR-CPET) was used to measure oxygen consumption and CO. Both SSc without PAH and SSc-PAH have a reduction of peak oxygen consumption. In SSc, it was due to a reduction of arteriovenous oxygen content difference (ΔAVo2), while in SSc-PAH, it was related to a reduction in ΔAVo2 and in cardiac index [105].

In conclusion, CPET can predict vascular disease in SSc, but RHC remains the gold standard for definitive PAH diagnosis [65]. Future research should focus on standardizing CPET protocols and establishing reference values specific to the SSc population (Table 1).

### 4.3. Stress Echocardiography (SE)

Stress echocardiography (SE) plays a critical role in assessing patients with confirmed or suspected PH [106]. Graded semi-supine exercise stress echocardiography is the preferred method. This approach is favored because PAsP quickly returns to baseline following exercise, decreasing by as much as 25% within 3-5 min, so Doppler measurements should be taken within one minute of test completion [106,107]. Alternatively, a hypoxic challenge serves as a viable stress test, particularly for patients at risk of PH or high-altitude pulmonary edema (HAPE). In this test, patients are stationary, which simplifies imaging [108].

During stress testing, images and Doppler readings should be taken at all stages, with contrast agents enhancing RV imaging and TR signals [109]. Key parameters include TR velocity, RV size and function, lateral annular tissue Doppler, and free wall systolic strain [110].

Since hypoxia can cause PA vasoconstriction, oxygen saturation should also be measured and reported during Doppler stress tests assessing PAP [111]. It is important to note that PAP is influenced by flow-dependent factors, meaning it can increase in conditions like anemia, hyperthyroidism, or physical exercise.

To ensure diagnostic accuracy, Doppler signals should be optimized by examining various angles, and the administration of agitated saline or contrast may improve TR Doppler signal quality. Additionally, care must be taken to avoid misinterpreting noise as Doppler signals to prevent pressure overestimation [112].

Recent evidence shows that SE can help identify individuals at risk or in the early stages of PAH development. Assessing PVR may provide a more sensitive marker. A steeper increase in PVR during stress suggests a higher risk for PAH [113]. In patients with PH, PAP increases even with mild activity. However, the clinical value of using SE to assess PAP in patients with resting PH is not well established [114]. The degree of PAP elevation during exercise does not offer clear prognostic value, as outcomes in PH are primarily determined by the right ventricle’s (RV) ability to handle the increased load [115]. This has shifted focus to RV function evaluation, especially through advanced techniques like 2D speckle-tracking strain and 3D imaging [116,117].

Since right heart failure (HF) typically appears late in the disease, there is potential value in measuring RV contractile reserve during stress testing. The inability to increase PAP during exercise, likely reflecting poor RV contractile reserve, is linked with worse outcomes [118].

Studies demonstrate that patients with PAH exhibit reduced RV contractile reserve during low-dose dobutamine SE, correlating with impaired exercise capacity [119].

Multicenter research, such as the RIGHT-NET study, confirms SE reliability in assessing right heart function, provided standardized protocols are followed [120].

Importantly, exercise echocardiography may help identify at-risk populations, such as SSc patients, where an abnormal rise in pulmonary artery pressure (>20 mmHg) during exercise strongly predicts future PH development [121]. Also, Codullo et al. demonstrated that an abnormal increase in PAsP during exercise was independently predictive of the development of PH in SSc patients. Their study suggests that this technique could serve as an effective screening tool for identifying patients at risk of PH, even in the absence of clinical signs [122].

Comparative studies suggest that noninvasive measures like SE and exercise cardiac MRI (ExCMRip) can detect early hemodynamic abnormalities, even in asymptomatic mutation carriers or post-treatment patients. These findings support SE’s role in early detection and functional assessment of PH. While echocardiography reasonably predicted these abnormal responses, it overestimated the slopes compared to CMRip and could only recover high-quality signals in 63% of patients during peak exercise. Additionally, RV contractility, which was higher at rest in PH patients, did not increase during exercise, indicating impaired adaptive response. The study concluded that while echocardiographic estimates of RV and pulmonary function are feasible during exercise and can detect pathology, some limitations remain to be addressed for broader clinical application [123] (Table 1).

## 5. Right Heart Catheterization and Endomyocardial Biopsy

The definition of PH is based on hemodynamic assessment by RHC. PH is defined as a condition characterized by mPAP value > 20 mmHg at rest [5]. In the definition of pre-capillary PH, it is essential to include RVP and pulmonary capillary wedge pressure (PAWP) values to discriminate elevated PAP due to pathology affecting the pulmonary vascular compartment from that due to left heart disease (LHD), increased pulmonary blood flow or intrathoracic pressure. Based on the available data, the upper limit of normality for RVP and the threshold value characterized by lower prognostic significance is approximately 2 Wood units (UW) [124]. In the SSc population, the group of patients with RVP > 2 UW had a worse prognosis [125]. For the definition of pre-capillary PH, it is recommended to consider a PAWP value ≤ 15 mmHg, bearing in mind that patient phenotype, risk factors, and echocardiographic findings such as left atrial volume should be taken into account when differentiating pre-capillary from post-capillary PH [126]. As it is, in any case, an invasive examination, eight tools have been proposed in order to be able to well direct a patient with SSc to perform RHC. They are clinical (progressive dyspnea in the last 3 months; unexplained dyspnea; worsening of the WHO functional class of dyspnea; any findings on subjective examination indicative of elevated right heart pressure; any signs of right heart failure), echocardiographic (PAsP > 45 mmHg; right ventricular dilatation) and obtained by pulmonary function tests (pulmonary carbon monoxide diffusing capacity < 50% without pulmonary fibrosis) In the assessment of early PVD, catheterization provides notable benefits compared to echocardiography or alternative techniques, as it permits highly accurate and precise measurement of mPAP, PAWP, CO, PVR and TPR, enabling the recognition of average PAP and PH values under stress, even if only slightly elevated [121].

Myocardial involvement in systemic sclerosis (SSc) is a significant factor in patient management and is linked to increased mortality. Due to the nonspecific and late presentation of symptoms [127,128], early detection is crucial. Endomyocardial biopsy serves as a valuable diagnostic tool, enabling the identification of myocardial necrosis, fibrosis, and vascular changes, even in the absence of coronary artery disease [129,130]. Histological analysis can reveal fibrosis in the myocardium and sinoatrial node, with greater fibrotic extension in SSc patients compared to healthy individuals [131]. This finding is associated with heart failure and poor prognosis [132]. Additionally, studies have shown reduced sarcomere function in SSc patients with pulmonary arterial hypertension (PAH), correlating with right ventricular dysfunction and adverse outcomes. Notably, sarcomere alterations are detectable even before PAH develops, suggesting a role in early diagnosis [130]. Immunofluorescence studies also indicate perisarcolemmal IgM deposits in SSc patients, even when cardiac symptoms are minimal [133].

A study by Milesi-Lecat et al. found that SSc patients undergoing endomyocardial biopsy exhibited myocardial perfusion abnormalities, with fibrosis present in some cases. Immunohistochemical analysis revealed an aberrant expression of HLA-DR on cardiac fibroblasts [134]. A subsequent study confirmed myocardial perfusion defects without coronary occlusion in some patients, while others had fibrosis linked to coronary spasms, along with altered HLA class II antigen expression [135].

Although cardiac MRI (CMR) and echocardiography can detect fibrosis [128], endomyocardial biopsy remains the gold standard for diagnosing myocardial involvement in SSc. It is particularly recommended in cases of ventricular arrhythmias, myocarditis, or pericarditis presenting as acute heart failure [136,137]. Additionally, histological findings correlate with PCR and CMR results in autoimmune patients, reinforcing the biopsy’s role in early differential diagnosis [138].

It can help to identify early forms of cardiac involvement before they become evident on non-invasive methods and before clinical manifestations [130,139]. Myocarditis in patients with SSc has clinical and histological features unfavorable compared to other forms. They more frequently present signs and symptoms of decompensation and greater fibrosis extension. The fibrosis extension does not differ significantly based on disease duration, suggesting that it develops early in the course of the disease. The extent of fibrosis correlates with the degree of skin involvement, the severity of right ventricular dysfunction on CMR, and ventricular arrhythmias on a 24 h Holter ECG. The extent of inflammatory damage is also a determinant of unfavorable outcomes. In a study by De Luca et al., some patients exhibited post-capillary pulmonary hypertension, likely secondary to myocardial involvement [140]. As well as in patients with SSc-PAH, fibrosis and inflammation have also been found in SSc symptomatic patients with mildly reduced left ventricular ejection fraction. Also, in these patients, the extension of cardiac fibrosis and inflammation was associated with poor prognosis [132]. Restrictive cardiomyopathy can be a manifestation of SSc, and endomyocardial biopsy is often necessary for differential diagnosis from other forms [141]. Differential diagnosis is essential because the diagnosis of cardiomyopathy and systemic sclerosis appears to be a poor prognostic indicator compared to the diagnosis of idiopathic dilated cardiomyopathy (DCM) [142]. Endomyocardial biopsy can be performed during heart catheterization; it is generally safe but may carry some complications. However, the diagnostic benefits outweigh the potential risks. In addition, not all areas of the heart present fibrosis, and it is therefore not always possible to detect it [130] (Table 1).

## 6. Endothelial Function

Microvascular and macrovascular damage are central to the pathogenesis of SSc and contribute to its various clinical manifestations. Endothelial dysfunction plays a pivotal role in driving these vascular abnormalities, leading to impaired vascular tone, inflammation, and fibrosis. In particular, PAH is driven by endothelial dysfunction, abnormal smooth muscle cell activity, and fibroblast dysregulation, resulting in vasoconstriction, microthrombosis, vascular fibrosis, and remodeling. Disruptions in endothelial function, including impaired growth factor synthesis and cell survival, further contribute to PAH progression. Additionally, the endothelial glycocalyx, a protective gel-like layer regulating vascular permeability, inflammation, and shear stress transmission, is crucial for maintaining vascular integrity. Its degradation has been implicated in the development of pulmonary hypertension, highlighting its role in disease progression [143].

Endothelial function is assessed through flow-mediated dilation (FMD) of the brachial or radial artery [144]. This non-invasive ultrasound-based method evaluates endothelium-dependent vasodilation as a marker of vascular health.

A sphygmomanometer cuff is placed distal to the artery, typically on the forearm, in cases involving the humeral artery. The assessment begins with a baseline diameter measurement of the artery after one minute of flow image acquisition. The cuff is then inflated to a suprasystolic pressure, at least 50 mmHg above systolic pressure, for five minutes to induce transient ischemia. Upon deflation, reactive hyperemia is evaluated by tracking changes in arterial diameter.

Reactive hyperemia will be assessed by calculating the ratio of the change in artery diameter (maximal dilation post-deflation minus baseline) to the baseline value, representing the peak FMD recovery. FMD will be quantified as the percentage increase in arterial diameter following a pressure stimulus.

Corrado et al. examined endothelial dysfunction in SSc by evaluating FMD and serum markers of vascular damage. SSc patients exhibited reduced FMD, a delayed peak response, and increased levels of VEGF, VCAM-1, and angiopoietin-2 compared to healthy controls. FMD impairment was associated with disease severity, including pulmonary hypertension and digital ulcers, and correlated with microvascular damage [145].

FMD, combined with echocardiography, could be a useful method to distinguish SSc patients from those with PAH. Correale et al. found that SSc-PAH patients show lower FMD values and impaired right ventricular function, with markers such as TAPSE < 18 mm and TRV > 280 cm/s. Lower FMD% predicts adverse cardiovascular outcomes, highlighting its potential role in early PAH detection. Identifying endothelial dysfunction and RV impairment may help predict PAH progression in SSc patients [12].

A study conducted on patients with PAH, including a subset with SSc, found that lower FMD values were significantly associated with higher PAsP and showed a borderline correlation with PVR. Patients classified by peak TRV and FMD levels into three groups (neither, either, or both impaired) exhibited a stepwise increase in PAsP, mPAP, and PVR. These findings suggest that FMD, alone or combined with peak TRV, serves as a noninvasive marker of pulmonary hypertension severity [146] (Table 1).

## 7. Future Directions

Xenon lung scanning is a safe, simple, accurate, and sensitive method for the early diagnosis of inhalation injury, with important prognostic implications. It may also aid in the diagnosis of PH. Combining xenon-enhanced ventilation with iodine-enhanced perfusion DECT allows for a comprehensive evaluation of pulmonary ventilation and perfusion. However, xenon-enhanced DECT has several limitations that have so far prevented its widespread use in clinical practice. With further advancements, this technique is expected to play an increasingly important role in the diagnosis and assessment of selected pulmonary abnormalities [147].

Xenon-129 gas-exchange magnetic resonance imaging/magnetic resonance spectroscopy (^129^Xe MRI/MRS) is an emerging functional lung imaging tool with the potential to provide precise, quantitative metrics of pulmonary ventilation, interstitial membrane uptake, transfer to capillary red blood cells (RBCs) and hemodynamics [148]. ^129^Xe MRI/MRS metrics can be used to differentiate between patients with COPD, idiopathic pulmonary fibrosis (IPF), PAH, and post-capillary PH [148]. The incorporation of cardiogenic ^129^Xe-RBC signal oscillations and RBC transfer defects into an initial diagnostic algorithm revealed good sensitivity and specificity in differentiating between pre- and post-capillary PH [149]. However, it remains unclear whether these ^129^Xe-RBC metrics are sensitive to changes associated with PAH-targeted therapies. Widespread clinical implementation is still a long way off.

To the best of our knowledge, there is currently no agreement on how these new methods will fit in with classic methods for the diagnosis of PH.

Photon-counting detector computed tomography (PCD-CT) iodine maps of the lung parenchyma, combined with perfusion scintigraphy for detecting and estimating the extent of pulmonary perfusion defects, appear to be a promising tool. Recently, PCD-CT iodine maps have demonstrated accurate detection of CTEPH, showing comparability to perfusion SPECT/CT with good quantitative correlation. Furthermore, the extent of visually assessed PCD-CT perfusion defects has been associated with prognostic right heart catheterization measurements [150].

## 8. Conclusions

The evaluation of the pulmonary circulation in patients with SSc is important because it allows the diagnosis of complications of SSc such as PH. Being able to predict the development of PH in these patients with non-invasive tests such as echocardiography, stress echocardiography, CPET, and MRI would allow us to save resources and avoid complications related to the invasive approach. Furthermore, by regularly subjecting patients with SSc to these non-invasive methods, the diagnosis of PAH could be anticipated and the use of specific drugs for PAH could be earlier, also improving the prognosis of patients with SSc.

## Figures and Tables

**Figure 1 diagnostics-15-01029-f001:**
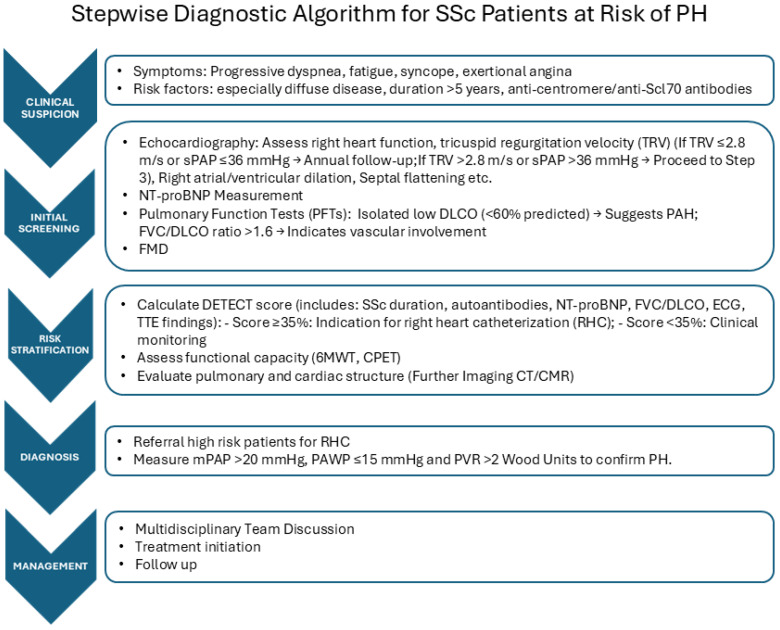
Right heart catheterization (RHC) represents the gold standard tool to identify different forms of PH, as it ensures the correct assessment of the hemodynamic parameters and is required to define the diagnosis of PAH [5]. Early detection of RV impairment, even in asymptomatic patients, is crucial to identify SSc subjects at risk of developing PH and to permit a prompt diagnosis of this condition, allowing one to start the appropriate treatment as soon as possible, as early treatment is associated with a better vital prognosis and improvement of quality of life. Nevertheless, as RCH is an invasive method for measuring cardiac pressure, in clinical practice, the availability of more accessible and less invasive, reliable diagnostic tools with non-significant costs is of crucial importance to identify patients that should have to undergo RHC to permit early diagnosis and to allow monitoring patients with defined PH. Further, RCH does not provide essential information about the causes of RV impairment, which can be due to myocardial inflammatory damage, pericardium involvement, or defect of conduction, and should be detected by using alternative diagnostic methods.

**Figure 2 diagnostics-15-01029-f002:**
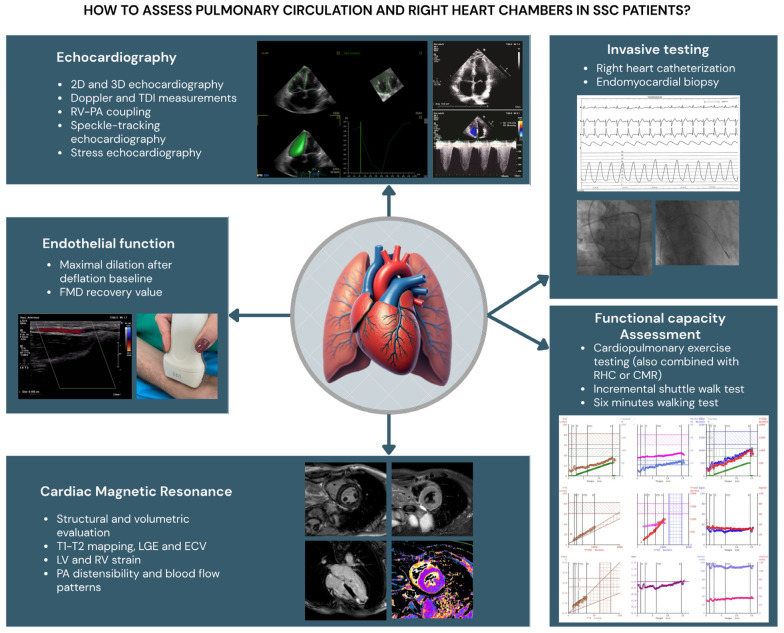
Diagnostic tools in clinical practice for the study of pulmonary circulation and right heart chambers tools in SSc patients. RV: right ventricle; LV: left ventricle; TAPSE: tricuspid annular plane systolic excursion; TDI: tissue Doppler imaging; PA: pulmonary artery; LGE: late gadolinium enhancement; ECM: extracellular volume; RHC: right heart catheterization; CMR: cardiac magnetic resonance; FMD: flow-mediated dilation.

**Figure 3 diagnostics-15-01029-f003:**
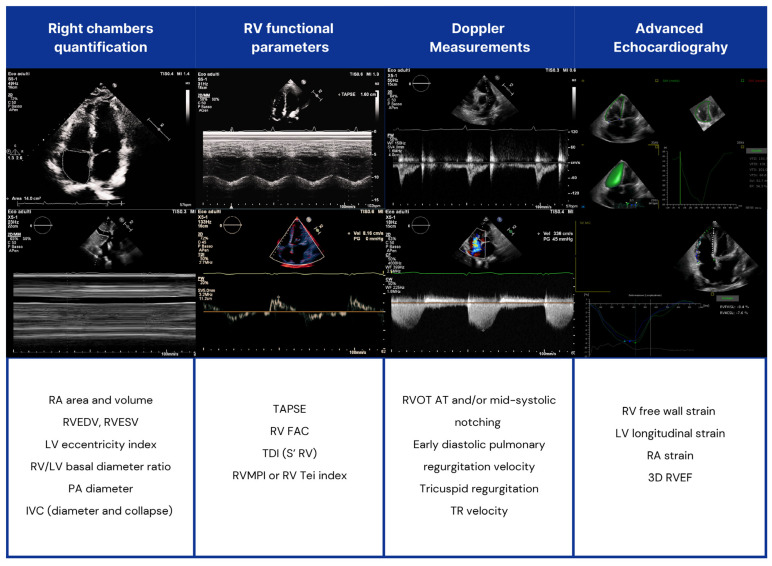
Echocardiographic assessment of right heart chambers in SSc patients. RV: right ventricle; LV: left ventricle; RVOT AT: RV outflow tract Doppler acceleration time; RA: right atrial; IVC: inferior vena cava; TAPSE: tricuspid annular plane systolic excursion; TDI: tissue Doppler imaging; PA: pulmonary artery; FAC: fractional area change; RVMPI: RV myocardial performance index; RVEDV: right end-diastolic volume (RVEDV); RVESV: end-systolic volume; RVEF: RV ejection fraction (EF).

**Table 1 diagnostics-15-01029-t001:** CMR: cardiac magnetic resonance; CPET: cardiopulmonary exercise testing; DLCO: diffusion capacity for carbon monoxide; FVC: forced vital capacity; RHC: right heart catheterization; SSc: systemic sclerosis.

Methods	Deviation for SSc from the Usual PAH Assessment	Reference
Resting echocardiography	Resting echocardiography is recommended as a screening test in asymptomatic patients with SSc, followed by annual screening with echocardiography, DLCO, and biomarkers	[5]
DETECT algorithm	In adult patients with SSc of >3 years disease duration, an FVC ≥ 40%, and a DLCO < 60%, the DETECT algorithm is recommended to identify asymptomatic patients with PAH	[5]
RHC	In patients with SSc where breathlessness remains unexplained following non-invasive assessment, RHC is recommended to exclude PAH	[5]
Exercise echocardiography or CPET or CMR	In symptomatic patients with SSc, exercise echocardiography, CPET, or CMR may be considered to aid in the decision to perform RHC	[5]

**Table 2 diagnostics-15-01029-t002:** Methods for evaluating right heart and pulmonary circulation.

**Two-dimensional echocardiography**	PAsP (5,11,12–13,15); Peak TRV (5,12–13); VCI (diameter and collapse) (5,12); RV/LV basal diameter ratio (5); LV eccentricity index (5,13); RVOT AT and/or mid-systolic notching (5); RA area (5); Early diastolic pulmonary regurgitation velocity (5); Pulmonary artery to aortic root ratio > 1 and/or PA diameter > 25 mm (12); TAPSE (12–13,15); TDI (S’ RV), diastolic dysfunction (12,22–23); RV FAC (12); TAPSE/PAsP (12,30–32); RA, RV, and LV longitudinal strain (27–29); RVMPI or RV Tei index (24–26).
**Three-dimensional echocardiography**	RV end-diastolic volume (14,18); RV end-systolic volumes (18); RV ejection fraction (14,18); Stroke volume (18); RA area (19); TRV peak (14,19); VAC (21).
**CMR (T1 mapping, LGE, SSFP)**	Right end-diastolic volume (RVEDV) (35–36,41); End-systolic volume (RVESV) (35); Ejection fraction (EF) (35–36); Ejection volume (35); CO (23,35,39); RV stroke volume (RVSV) (23,36,39); RA volume (38); Ventricular mass index (43); Curvature of the interventricular septum (36); PAP (35); PVR (35); Parameters of the PA, including Flow variations, average and peak flow velocity, RAC, wedge pressure velocity, and elasticity of the PA (36–38); Myocardial perfusion (36); Fibrosis (36); Extracellular volume (42).
**ISWT**	Distance (47–48).
**6MWT**	Distance (49); Saturation (70).
**CPET**	VO2 peak (72,78–79,84–85,87–88,90,95–97) and AT (79,86,96-97); VCO2 (72); PETCO2 (79,95–97); VE (72); HR (72,95); VE/VCO2 slope (72,78–80,84,87–89,96–98); OUES (70,72,78); O2 pulse (79,86,91–92,98); WR peak and AT (70,95); VO2/WR relationship (70); VD/VT ratio (70); METS at peak exercise (85); Interval between AT and RCP (85); SaO2 (87,88).
**CPET with RHC (CPET/RHC)**	[P(Ai-a)O2] (98–99); Peak VO2, PETC02, VEint, VP, VEVCO2 (101–102).
**CMR-CPET**	Oxygen consumption (107); Cardiac Output (107).
**SE**	TR velocity (111,112); RV size and function (112); Lateral annular tissue Doppler (112); Systolic strain of the free wall (112); Size and collapsibility of the inferior vena cava; PAsP estimation (116–117,124); PVR estimation (115); RV function evaluation (118–119); RV contractile reserve estimation (121,125).
**RHC**	PAPm (123); RVP (123); PAWP (123); CO (123); Endomyocardial biopsy (129–144).
**FMD**	Baseline diameter (146–148); Maximaldilation after deflation baseline (146–148); FMD recovery value (153–148).

PAsP: pulmonary artery systolic pressure; TRV: tricuspid regurgitation velocity; IVC: inferior vena cava; RV: right ventricle; LV: left ventricle; RVOT AT: RV outflow tract doppler acceleration time; RA: right atrial; TAPSE: tricuspid annular plane systolic excursion; TDI: tissue doppler imaging; PA: pulmonary artery; FAC: fractional area change; PISA: proximal isovelocity surface area; EROA: effective regurgitant orifice area; RVMPI: RV myocardial performance index; VAC: Ventricular–Arterial Coupling; LGE: late gadolinium enhancement; SSFP: steady-state free procession; RVEDV: right end-diastolic volume (RVEDV); RVESV: end-systolic volume; EF: ejection fraction (EF); RVSV: RV stroke volume; PAP: pulmonary artery pressure; PVR: pulmonary vascular resistance; RAC: relative area change; ISWT: Incremental Shuttle Walking Test; 6MWT: six minutes walking test; CPET: cardiopulmonary exercise test; VO2: oxygen uptake; AT: anaerobic threshold; VCO2: carbon dioxide production; VE: minute ventilation; HR: heart rate; VE/VCO2: ventilatory equivalent for carbon dioxide; OUES: oxygen uptake efficiency slope; WR: work rate; VD: dead volume; VT: tidal volume; METS: Metabolic Equivalent of Task; RCP: respiratory compensation point; SaO2: oxygen saturation; [P(Ai-a)O2] exercise alveolar–arterial oxygen gradient; RHC: right heart catheterization; CMR: cardiac magnetic resonance; SE: stress echocardiography; PAPm: mean pulmonary artery pressure; PAWP: pulmonary artery wedge pressure; CO: cardiac output; Ees: end-systolic elastance; Ea: effective arterial elastance; FMD: flow-mediated dilation.

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
