# Peer review of "How to Assess Pulmonary Circulation and Right Heart Chambers in Systemic Sclerosis Patients?"

_diagnostics, 2025, doi:10.3390/diagnostics15081029_

Round 1
Reviewer 1 Report
Comments and Suggestions for Authors
Dear authors,
Congratulation for your valuable work.
It is a comprehensive review that addresses the complete panel of paraclinical investigations used for assesing pulmonary circulation in SSc patients.
Although you have specified the importance of DLCO, in your paper the approach is only from the PAH perspective. Lung and right heart damage in patients with SSc should also be mentioned, since interstitial lung damage is a frequent type of damage present in patients with scleroderma.and may also cause pulmonary hypertension. To fully evaluate the patient, I suggest grouping these investigations: high resolution CT, DLCO, and body pletismography.
Along with Six Minutes Walking Test and pulmonary exercise cardio tests, I would also recommend adding nocturnal pulse oximetry as a method of evaluation.
Author Response
Reviewer n.1
Dear authors,
Congratulation for your valuable work.
It is a comprehensive review that addresses the complete panel of paraclinical investigations used for assesing pulmonary circulation in SSc patients.
Although you have specified the importance of DLCO, in your paper the approach is only from the PAH perspective. Lung and right heart damage in patients with SSc should also be mentioned, since interstitial lung damage is a frequent type of damage present in patients with scleroderma and may also cause pulmonary hypertension. To fully evaluate the patient, I suggest grouping these investigations: high resolution CT, DLCO, and body pletismography.
Thank you very much for your comment. We have mentioned lung and right heart damage in SSc patients. As per your request we briefly discussed the following topics: HR-CT, DLCO and body pletismography.
Along with Six Minutes Walking Test and pulmonary exercise cardio tests, I would also recommend adding nocturnal pulse oximetry as a method of evaluation.
As per your request we added nocturnal pulse oximetry.
Reviewer 2 Report
Comments and Suggestions for Authors
Thank you for giving me an opportunity to review the article. The article provides a comprehensive overview of diagnostic tools for assessing pulmonary circulation and right heart function in systemic sclerosis (SSc) patients. Below are my comments.
1. Please include a glossray to define technical terms for more accessibility.
2. Please inculded a flowchat illustrating a stepwise diagnostic algorithm for SSc patients at risk of PH.
3. Expand on biomarkers (e.g.,NT-proBNP) and their integration with imaing findings for risk stratification.
4. Emphasize the importance of multidisciplinary Team (MDT) meetings including rheumatologists, pulmonologists, cardiologists, and radiologists.
Author Response
Reviever n. 2
Thank you for giving me an opportunity to review the article. The article provides a comprehensive overview of diagnostic tools for assessing pulmonary circulation and right heart function in systemic sclerosis (SSc) patients. Below are my comments.
1. Please include a glossary to define technical terms for more accessibility. We added the glossary
- Please included a flowchat illustrating a stepwise diagnostic algorithm for SSc patients at risk of PH.
We added the flowchart as per your request
Expand on biomarkers (e.g.,NT-proBNP) and their integration with imaging findings for risk stratification.
We expanded on biomarkers and their integration with imaging findings
Emphasize the importance of multidisciplinary Team (MDT) meetings including rheumatologists, pulmonologists, cardiologists, and radiologists.
In the text we emphasized the importance of multidisciplinary team meetings.
Reviewer 3 Report
Comments and Suggestions for Authors
The reviewed article evaluates a very important aspect related to systemic sclerosis management, meaning the involvement of pulmonary circulation, and right ventricle (RV) involvement. Early identification of RV dysfunction can lead to more effective treatment, reduced complications, and better patient survival. So, an article that could contribute to a deeper understanding of how diagnostic tools can be optimized for this specific patient population is very useful.
The text provides an extensive overview, detailing its diagnostic and prognostic utility. The inclusion of multiple studies strengthens its scientific credibility. The strengths and limitations of each of the methods are well articulated, helping to contextualize their role in clinical practice.
Some comments
- since the discussion it’s about pulmonary circulation involvement I think that the introductory part would benefit from some ethiopatogeny clues, including some well designed visual schemes. Also types of associated pulmonary hypertension should be discussed
- even if the article is very interesting it seems to me that the discussion starts too abruptly and is mainly around general ways of cardio-pulmonary evaluation, not necessarily directed to particularities of systemic sclerosis patients
- please revise the order of the references since part of them do not follow the order (see especially Table 1)
- Some sentences are overly long and complex, making readability difficult
- it could be slightly refined for clarity, conciseness, and grammatical accuracy
- please verify that there is term consistency “CMRI” vs. “CMR” – The text switches between CMRI and CMR (Cardiac Magnetic Resonance Imaging). CMR is the standard abbreviation in cardiology literature—consider using it consistently.
- the discussion could be better structured to separate diagnostic capabilities, prognostic markers, and emerging research into clearer subsections
- While the text constantly cites important studies (e.g., Knight et al., Kobayashi et al.), it would benefit from briefly stating the clinical implications of these findings rather than just summarizing results.
- 30% iThenticate report is quite high, this should be revised
- text is understandable, but as a non-native speaker I'm not qualified to judge it, nevertheless some spelling errors should be corrected
Author Response
Reviever n. 3
The reviewed article evaluates a very important aspect related to systemic sclerosis management, meaning the involvement of pulmonary circulation, and right ventricle (RV) involvement. Early identification of RV dysfunction can lead to more effective treatment, reduced complications, and better patient survival. So, an article that could contribute to a deeper understanding of how diagnostic tools can be optimized for this specific patient population is very useful.
Thank you very much for your comment.
The text provides an extensive overview, detailing its diagnostic and prognostic utility. The inclusion of multiple studies strengthens its scientific credibility. The strengths and limitations of each of the methods are well articulated, helping to contextualize their role in clinical practice.
Some comments
- since the discussion it’s about pulmonary circulation involvement I think that the introductory part would benefit from some ethiopatogeny clues, including some well designed visual schemes. Also types of associated pulmonary hypertension should be discussed
We revised the introduction according to the reviewer’ suggestion (adding some ethiopatogeny clues) and we reported different types of PH in SSc.
- even if the article is very interesting it seems to me that the discussion starts too abruptly and is mainly around general ways of cardio-pulmonary evaluation, not necessarily directed to particularities of systemic sclerosis patients
We changed the discussion focusing on SSc patients.
- please revise the order of the references since part of them do not follow the order (see especially Table 1)
- Some sentences are overly long and complex, making readability difficult
We revised these long sentences.
- it could be slightly refined for clarity, conciseness, and grammatical accuracy
We revised the text according to the reviewer’suggestions
- please verify that there is term consistency “CMRI” vs. “CMR” – The text switches between CMRI and CMR (Cardiac Magnetic Resonance Imaging). CMR is the standard abbreviation in cardiology literature—consider using it consistently.
We used the standard abbreviation “CMR”
- the discussion could be better structured to separate diagnostic capabilities, prognostic markers, and emerging research into clearer subsections
- While the text constantly cites important studies (e.g., Knight et al., Kobayashi et al.), it would benefit from briefly stating the clinical implications of these findings rather than just summarizing results.
We tried to report the clinical implication of the study rather that summarizing results.
- 30% iThenticate report is quite high, this should be revised
The text was revised.
Comments on the Quality of English Language
- text is understandable, but as a non-native speaker I'm not qualified to judge it, nevertheless some spelling errors should be corrected
Reviewer 4 Report
Comments and Suggestions for Authors
In this highly comprehensive review for assessment modalities for the right heart, Dr Correale provided a a holistic summary for PH in setting of SSc. A few points could be improved
- Consider a figure / table, highlighting any deviation for SSc from the usual PAH assessment as outlined by 2022 ESC / ERS guideline.
- Distill and synthesize your message. For example, from Line 506 till 523 were merely talking about individual study. Consider synthesize into one brief paragraph before audience (likely junior pulmonologist / cardiologist as opposed to baasic scientist) lose track
- How would new modalities be integrated into clinical practice in the long run - think abt Xenon lung scan for PAH, photon counting iodine maps etc.
Author Response
Reviever n. 4
In this highly comprehensive review for assessment modalities for the right heart, Dr Correale provided a a holistic summary for PH in setting of SSc. A few points could be improved
- Consider a figure / table, highlighting any deviation for SSc from the usual PAH assessment as outlined by 2022 ESC / ERS guideline.
We report a table with any deviation for SSc patients from the usual PAH assesment.
- Distill and synthesize your message. For example, from Line 506 till 523 were merely talking about individual study. Consider synthesize into one brief paragraph before audience (likely junior pulmonologist / cardiologist as opposed to baasic scientist) lose track
We revise the text, synthesizing the sentences.
- How would new modalities be integrated into clinical practice in the long run - think abt Xenon lung scan for PAH, photon counting iodine maps etc.
We added a new small chapter called “Future Directions” in order to answer the comment of the reviewer.
Round 2
Reviewer 3 Report
Comments and Suggestions for Authors
Just minor spelling mistakes. Also, 30% iThenticate report is still high. Please verify!
Author Response
We have corrected the text as suggested by the reviewer. and we have tried to reduce any similarities with other papers.